EMBO
Molecular Medicine

# PERK activation mitigates tau pathology *in vitro* and *in vivo*

Julius Bruch[1,2,3,†], Hong Xu[1,2,3,†], Thomas W Rösler[1,2,†], Anderson De Andrade[1,3], Peer-Hendrik Kuhn[4,5], Stefan F Lichtenthaler[3,4,6], Thomas Arzberger[7], Konstanze F Winklhofer[3,8], Ulrich Müller[9] & Günter U Höglinger[1,2,3,*] (ID)

## Abstract

The RNA-like endoplasmic reticulum kinase (PERK) is genetically associated with the tauopathy progressive supranuclear palsy (PSP). To elucidate the functional mechanisms underlying this association, we explored PERK activity in brains of PSP patients and its function in three tauopathy models (cultured human neurons overexpressing 4-repeat wild-type tau or treated with the environmental neurotoxin annonacin, and P301S tau transgenic mice). *In vitro*, treatment with a pharmacological PERK activator CCT020312 or PERK overexpression reduced tau phosphorylation, tau conformational change and 4-repeat tau isoforms, and increased cell viability. *In vivo*, the PERK activator significantly improved memory and locomotor function, reduced tau pathology, and prevented dendritic spine and motoneuron loss in P301S tau mice. Importantly, the PERK substrate EIF2A, mediating some detrimental effects of PERK signaling, was downregulated in PSP brains and tauopathy models, suggesting that the alternative PERK–NRF2 pathway accounts for these beneficial effects in the context of tauopathies. In summary, PERK activation may be a novel strategy to treat PSP and eventually other tauopathies.

**Keywords** EIF2A; NRF2; PERK; progressive supranuclear palsy; tauopathy
**Subject Categories** Neuroscience; Pharmacology & Drug Discovery

## Introduction

Tauopathies are neurodegenerative diseases caused by aggregation of the microtubule-associated protein tau, encoded by the gene *MAPT*. Progressive supranuclear palsy (PSP) is a sporadic tauopathy, characterized by predominant involvement of tau isoforms with four microtubule-binding repeats (4R tau; Chambers *et al*, 1999). In a genome-wide association study, we identified common variants of *EIF2AK3,* encoding the RNA-like endoplasmic reticulum kinase (PERK), to increase risk for PSP (Höglinger *et al*, 2011). There is also neuropathological evidence suggesting involvement of abnormal PERK signaling in tauopathies (Nijholt *et al*, 2012; Stutzbach *et al*, 2013). However, the exact molecular mechanisms of PERK function in PSP are not yet understood.

PERK is an integral part of the unfolded protein response (UPR). An overload of misfolded proteins in the endoplasmic reticulum (ER stress) activates PERK which subsequently phosphorylates the eukaryotic translation initiation factor 2-alpha (EIF2A; Schroder & Kaufman, 2005) and the nuclear factor erythroid 2-related factor 2 (NRF2; Cullinan *et al*, 2003), resulting in global suppression of translation and transcription of cytoprotective factors. Cellular tau accumulation may also activate PERK (Abisambra *et al*, 2013). PERK, in turn, activates GSK3ß, a tau kinase (Baltzis *et al*, 2007; Nijholt *et al*, 2013).

Pharmacological PERK inhibition has been reported to reduce neurodegeneration in P301L tau transgenic mice (Radford *et al*, 2015). The rationale behind PERK inhibition was that chronic overactivation of PERK in tauopathies may lead to a disadvantageous long-term suppression of translation by phosphorylation of EIF2A (Radford *et al*, 2015). However, in brain sections of Wolcott–Rallison syndrome, resulting from PERK deficiency due to *EIF2AK3* loss-of-function mutations, we identified tau-positive neurofibrillary tangles (Bruch *et al*, 2015). Furthermore, phosphorylated tau has been shown to decrease upon pharmacological activation of NRF2 (i.e., the second major PERK target), and to increase upon knockout of NRF2 (Jo *et al*, 2014).

1  Department of Translational Neurodegeneration, German Center for Neurodegenerative Diseases (DZNE), Munich, Germany
2  Department of Neurology, Technical University of Munich (TUM), Munich, Germany
3  Munich Cluster for Systems Neurology (SyNergy), Munich, Germany
4  Neuroproteomics, Klinikum rechts der Isar and Institute for Advanced Study, Technical University of Munich (TUM), Munich, Germany
5  Institute of Pathology, Technical University of Munich (TUM), Munich, Germany
6  Neuroproteomics, German Center for Neurodegenerative Diseases (DZNE), Munich, Germany
7  Center for Neuropathology and Prion Research (ZNP), University of Munich, Munich, Germany
8  Department of Molecular Cell Biology, Institute of Biochemistry and Pathobiochemistry, Ruhr University, Bochum, Germany
9  Institute for Human Genetics, University of Giessen, Giessen, Germany
  *Corresponding author. Tel: +49 89 4400 46464; E-mail: guenter.hoeglinger@dzne.de
  †These authors contributed equally to this work

Thus, the available data on the functional consequences of PERK signaling in tauopathies are inconclusive. We therefore investigated the state of PERK, EIF2A, and NRF2 in postmortem human PSP brains and the effects and mechanisms of PERK activation and inhibition in models of tauopathies.

# Results

### PERK is upregulated in PSP, but its substrates EIF2A and NRF2 behave differently

Western blot analyses were performed to assess the state of the UPR in human frontal cortex that is consistently involved in PSP (Fig 1A and B). An increase in phosphorylated PERK (pPERK) was detected in PSP, as previously shown by immunohistochemistry (Nijholt *et al*, 2012; Stutzbach *et al*, 2013). Total PERK protein was increased as well. Surprisingly, the PERK substrate EIF2A and phosphorylated EIF2A (pEIF2A) were reduced in PSP. However, the second major PERK substrate, NRF2, was increased with regard to total and phosphorylated protein (pNRF2). The ratios of pPERK/PERK, pNRF2/NRF2, and pEIF2A/EIF2A were not significantly altered. These findings suggest that PERK and NRF2 signaling are increased, and EIF2A signaling is decreased in PSP. It needs to be stressed, however, that the degree of phosphorylation is volatile in postmortem brain tissue even if it was frozen immediately after death (Wang *et al*, 2015).

### UPR changes in *in vitro* models of tauopathy

We then investigated the state of the UPR in two cellular models of PSP using LUHMES neurons. LUHMES neurons are differentiated from fetal human neuroblasts (Lotharius *et al*, 2005), to avoid limitations of rodent and tumor neuronal cell lines. Thapsigargin induces UPR stress and served as positive control for PERK and EIF2A phosphorylation (Fig 1C and D).

We used annonacin, as an environmental model of PSP, because its consumption causes the PSP-like tauopathy in Guadeloupe (Lannuzel *et al*, 2007). Cultured neurons treated with low doses of annonacin exhibit hallmarks of tauopathy, including increased tau concentration, tau hyperphosphorylation, cell death and somatodendritic redistribution of hyperphosphorylated tau in a manner dependent on mitochondrial complex I activity (Lannuzel *et al*, 2003; Escobar-Khondiker *et al*, 2007; Yamada *et al*, 2014). Annonacin (25 nM for 48 h) increased pPERK, while total PERK and EIF2A were reduced (Fig 1C and D). Although we only tested selective phospho-tau residues (AD2, CP13, AT180, AT8), these represent the overall state of tau phosphorylation in PSP (Feany *et al*, 1995; Wray *et al*, 2008).

Next, we lentivirally expressed either wild-type 2N4R tau, wild-type 2N3R tau, or the fluorescent control protein mCherry (Shaner *et al*, 2004) under control of a ubiquitin promoter. The 2N4R tau-encoding virus increased 4R tau mRNA from $80 \pm 10$ (control) to $5,000 \pm 2,000$ copies/ng total mRNA; the 2N3R tau-encoding virus increased 3R tau mRNA from $1,800 \pm 700$ to $7,600 \pm 300$ copies/ng total mRNA; the specific increase of 4R and 3R tau protein was

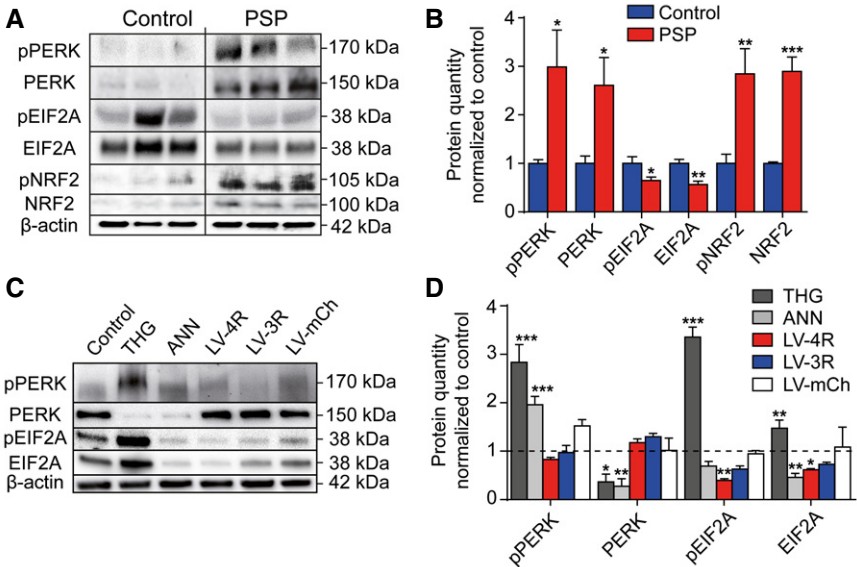

**Figure 1.  Unfolded protein response in PSP and tauopathy models.**

A   Representative Western blots of gyrus frontalis superior homogenates of controls without neurodegenerative disorder and PSP patients.
B   Densitometric analysis of Western blots described in (A) (controls, *n* = 6; PSP, *n* = 7).
C   Representative Western blots of protein extracts from LUHMES neurons after 48-h treatment with thapsigargin (THG, 30 nM), an inducer of ER stress (positive control), or annonacin (ANN, 25 nM), or after lentivirus-induced overexpression of 4R tau (LV-4R) or 3R tau (LV-3R).
D   Densitometric analysis of Western blots described in (C).

Data information: Results were normalized to untreated control neurons set as 1 (dashed line). Data are mean + SEM (*n* = 3 per condition). Statistical analysis in (B) was Student's *t*-tests and in (D) one-way ANOVA with Dunnett's *post hoc* test. *P < 0.05, **P < 0.01, ***P < 0.001 versus control.

confirmed by Western blot (Appendix Fig S1A and B). For both 4R and 3R tau, this means a significant concentration increase relative to control concentration. However, 10-fold concentration variations are also seen naturally (Mangin *et al*, 1989). None of the lentiviral models significantly increased PERK or pPERK, showing that tau overload *per se* is not sufficient to activate the UPR. However, similar to the situation in PSP, 4R tau (but not 3R tau or mCherry) suppressed EIF2A at the protein level. pEIF2A was reduced by both 4R and 3R tau.

Thus, both annonacin and wild-type 4R tau overexpression mimic the EIF2A and pEIF2A downregulation seen in PSP. However, none of these models showed significant upregulation of PERK as seen in PSP.

In order to understand whether PERK upregulation might only occur in the longer term, we compared 2- and 6-month-old P301S tau transgenic mice to controls (Fig EV1). Indeed, pEIF2A was downregulated in 2- and 6-month-old mice, but PERK, pPERK, and pNRF2 were only increased in 6-month but not in 2-month-old mice.

**Pharmacological modulation of the UPR *in vitro***

While a few specific PERK inhibitors are available (Wang *et al*, 2010; Axten *et al*, 2012; Atkins *et al*, 2013), the PERK activator CCT020312 is unique so far, because it does not work by inducing ER stress, but instead selectively activates PERK signaling (Stockwell *et al*, 2012). We tested the PERK inhibitor GSK2606414 (Wang *et al*, 2010; Axten *et al*, 2012; Atkins *et al*, 2013) and the PERK activator CCT020312 (Stockwell *et al*, 2012) in LUHMES neurons.

First, non-toxic concentrations were identified that effectively modify EIF2A phosphorylation (PERK activator: 200 nM; PERK inhibitor: 300 nM; Fig EV2A–C). These concentrations were used in all further experiments.

In the presence of annonacin (Fig EV2D and E), the PERK activator increased pNRF2, but not pEIF2A; the PERK inhibitor reduced pEIF2A levels significantly.

When overexpressing 4R tau in neurons, PERK activator treatment still increased pNRF2, but not pEIF2A (Fig EV2F and G), while PERK inhibitor treatment decreased pEIF2A but not pNRF2.

Upon 3R tau overexpression, PERK modulation showed no significant changes.

**PERK activation protects neurons *in vitro***

Next, we tested the effect of PERK modulation on the viability of LUHMES neurons in the MTT assay and by measurement of intracellular ATP concentrations.

Annonacin caused a concentration-dependent decline in both MTT signal and ATP concentration (Fig 2A and B). Co-treatment with the PERK inhibitor did not improve the cell metabolic activity (MTT signal). Only ATP concentrations improved at the lowest annonacin concentration (12.5 nM). However, the PERK activator improved both metabolic activity and ATP concentrations over a broad range of annonacin concentrations.

In the tau overexpression model, only 4R tau but not 3R tau caused significant neurotoxicity compared to the control protein mCherry (Fig 2C and D). The PERK activator reduced 4R tau toxicity as determined in MTT and ATP assays. The PERK inhibitor was slightly protective in the ATP assay only.

Protection of cells by the PERK activator was also seen morphologically. Annonacin and 4R tau overexpression caused chromatin clumps (Fig 2E) and reduced the density of the neuritic network (Fig 2F; quantified in Fig EV3). Counts of viable cells (Fig 2G) confirmed that the PERK activator mitigated the toxicity of both annonacin and 4R tau overexpression.

Next, we asked whether the protective effect of the PERK activator would be mediated via the NRF2 signaling arm. NRF2 regulates several downstream pathways, including iron sequestration controlled by heme oxygenase-1 (HO-1; Gorrini *et al*, 2013). We found a significant increase in HO-1 expression in neurons treated with the PERK activator (Fig EV4A and B), which indicates an activation of the PERK–NRF2 pathway. We then wanted to confirm whether modulation of NRF2 signaling would also affect the neuroprotective efficacy of the PERK activator in our lentiviral 4R tau overexpression model. Indeed, the toxicity of LV-4R was significantly reduced by treatment with an activator of NRF2 (DL-sulforaphane *N*-acetyl-L-cysteine; Fig EV4C), and exacerbated by siRNA targeting *NFE2L2* (the NRF2 gene; Fig EV4D and Appendix Fig S4; Dinkova-Kostova *et al*, 2002; Wang *et al*, 2012).

These findings suggest that the PERK activator protects neurons in two independent tauopathy models via the PERK–NRF2 pathway. Therefore, we proceeded to address the effect of the PERK activator on the tau protein in these models.

**PERK activation reduces pathological tau conformation and phosphorylation in the annonacin model**

The MC1 antibody detects conformationally altered tau as marker of early pathology (Jicha *et al*, 1997). Annonacin but not 4R or 3R tau overexpression increased the level of MC1-immunoreactive tau in LUHMES neurons. This effect was reversed by PERK activator treatment (Fig 3A and B) and was confirmed on dot blots with non-denatured proteins (Fig 3C).

Annonacin also increased phosphorylation of tau on the epitopes Ser-202 (CP13 antibody; Ishizawa *et al*, 2003) and Ser-396 and Ser-404 (AD2 antibody; Buee-Scherrer *et al*, 1996; Fig 3A and B). The PERK activator reduced this effect.

**PERK activation reduces the 4R tau isoform shift in the annonacin model**

Since the tau 3R/4R ratio is shifted toward the 4R isoform in PSP and some other tauopathies (Chambers *et al*, 1999) for reasons only partially understood (Bruch *et al*, 2014), we studied this isoform shift in the annonacin model.

Total tau protein (HT7 antibody) was not altered by treatment with annonacin or the PERK activator (Fig 3A and D). Consistent with previous work (Bruch *et al*, 2014), annonacin increased 4R tau, but not 3R tau levels, both at the protein and mRNA level (Fig 3A, D and E). The PERK activator reduced the amount of 4R tau mRNA and protein, and almost completely reversed the effect of annonacin on the 3R/4R ratio (Fig 3A, D and E).

In the presence of annonacin, mRNA levels of the tau splicing factor *SRSF2* increased (Fig 3F). Interestingly, the PERK activator blocked this effect. Other tested splicing factors causing alternative 3R and 4R tau splicing were not affected (Liu & Gong, 2008).

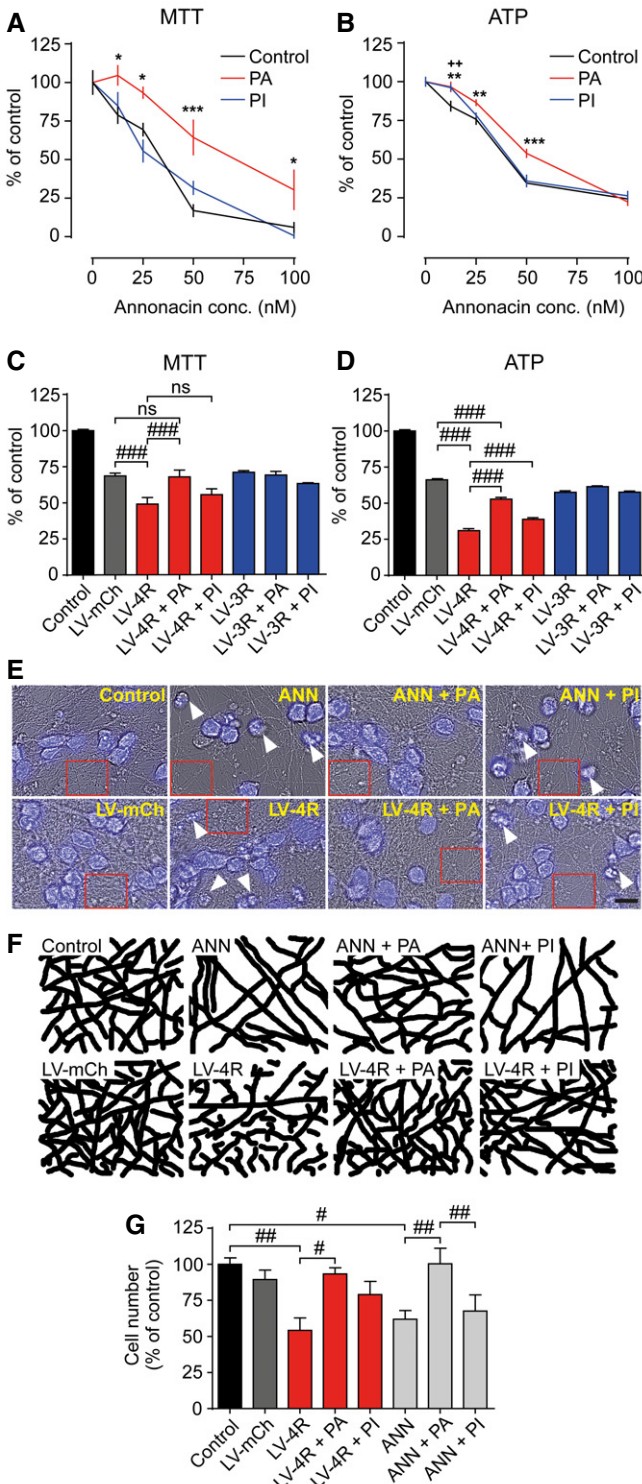

Figure 2.  PERK activator, but not inhibitor, is neuroprotective *in vitro*.

A, B  MTT viability assay and ATP concentrations in LUHMES neurons treated with different concentrations of annonacin and either no treatment (control) or PERK activator (PA, 200 nM) or PERK inhibitor (PI, 300 nM; n = 12 per condition).

C, D  MTT assay and ATP concentrations in LUHMES neurons either untreated (control) or transduced with lentiviruses to overexpress mCherry (LV-mCh, a red fluorescent control protein), 4R tau (LV-4R), or 3R tau (LV-3R), and either PI or PA treatment (n = 12).

E, F  Bright-field (gray) and overlaid DAPI images (blue) of LUHMES neurons left untreated (control) or treated with annonacin (ANN, 25 nM), or with LV-mCh or LV-4R, and PA or PI (E); scale bar: 100 μm; white arrows highlight dead neurons with condensed chromatin clumps. Chromatin clumps are condensations of chromatin, which stain deeply with DAPI. They are a typical microscopic feature of degenerated neurons (Pannese, 2015). Red rectangles mark the areas selected to show the neuritic network, displayed in detail in (F).

G  Numbers of viable neurons with intact nuclear and somatic morphology under the conditions described in (E) (n = 4).

Data information: Data are mean + SEM. Statistical analysis in (A and B) was two-way ANOVA with Dunnett's (A) or Holm–Sidak's (B) *post hoc* test; *$P < 0.05$, **$P < 0.01$, ***$P < 0.001$ PA versus control; ++$P < 0.01$ PI versus control. Statistical analysis in (C, D and G) was one-way ANOVA with Tukey's *post hoc* test. ns: not significant, #$P < 0.05$, ##$P < 0.01$, ###$P < 0.001$.

## PERK activation prevents annonacin-induced neurofilament dephosphorylation

Neurofilaments are cytoskeletal proteins in axons. Their phosphorylation is essential for axonal functions. Dephosphorylation of the neurofilament medium polypeptide (NFM) impairs axonal calibers (Save *et al*, 2004). Abnormal phosphorylation of neurofilaments has also been associated with tauopathy (Roher *et al*, 2013). Western blots of LUHMES neurons treated with annonacin and the PERK activator showed that phosphorylated NFM was decreased by annonacin and the decline was prevented by PERK activator (Appendix Fig S2A and B). This suggests that axonal pathology induced by annonacin-treated cells was associated with dephosphorylation of NFL and prevented by PERK activator treatment.

## Protective effects of PERK overexpression

We aimed to confirm the effects of PERK activation with an independent method. Overexpression of wild-type PERK leads to its activation in the absence of activating signals, because levels of ectopic PERK exceed levels of available ER-resident chaperone binding immunoglobulin protein BiP, leading finally to EIF2A phosphorylation in the absence of ER stress (Bertolotti *et al*, 2000). Thus, we created a lentivirus overexpressing wild-type PERK threefold (Fig EV5A).

PERK overexpression reversed the annonacin-induced increase in 4R *MAPT* mRNA (Fig EV5B) and protected LUHMES neurons against toxicity induced by annonacin (Fig EV5C and D) and 4R tau overexpression in a similar fashion to PERK activator treatment (Fig EV5E and F). These data confirm that the effects observed with the pharmacological PERK activator are indeed mediated by PERK activation and not by off-target effects.

## PERK activator target engagement *in vivo*

We then explored the utility of the PERK activator *in vivo* in a well-characterized P301S tau transgenic mouse model (Allen *et al*, 2002).

## PERK activation reduces tau phosphorylation in the overexpression model

Cells transduced with 4R tau-overexpressing lentivirus showed increased levels in CP13-, AD2-, and HT7 tau (but not MC1). CP13- but not AD2- and HT7 tau were normalized by simultaneous PERK activator treatment (Fig 3G and H).

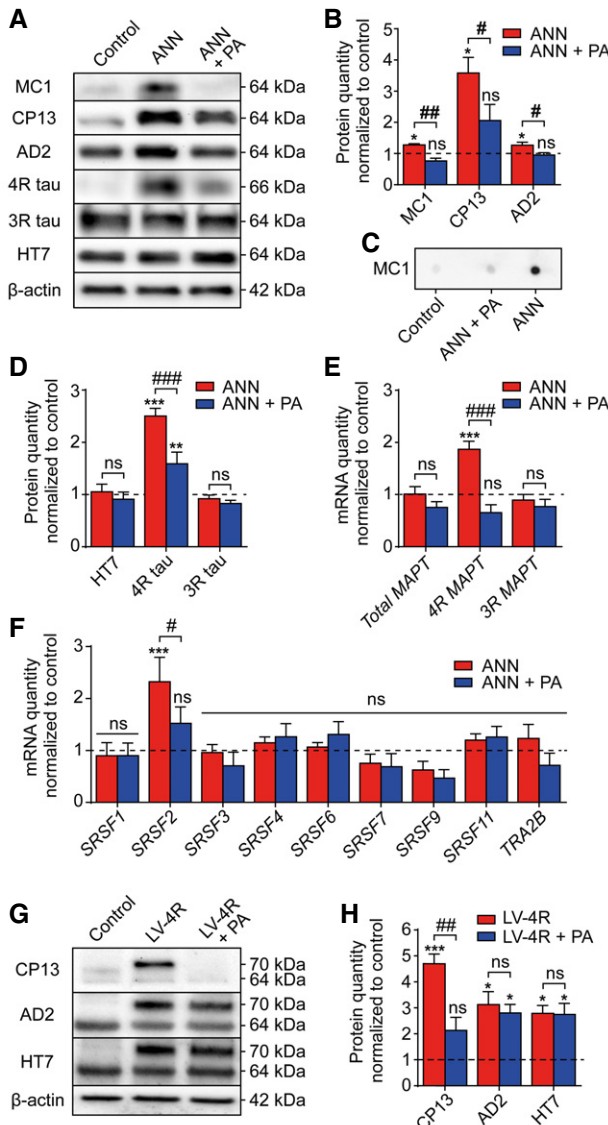

**Figure 3.   PERK activator decreases pathological tau species *in vitro*.**

A   Representative Western blots of protein extracts from LUHMES neurons left untreated (control) or treated with annonacin (ANN, 25 nM) in the absence or presence of PERK activator (PA, 200 nM). Antibodies depicted conformationally changed tau (MC1), phosphorylated tau (CP13; AD2), total tau (HT7), or 4R and 3R tau isoforms. Actin was used as loading control. The reason for the significant increase in 4R tau, without a remarkable change in total tau, is due to the relatively higher concentration of 3R tau in the LUHMES cells.

B   Densitometric analysis of Western blots described in (A) (*n* = 3 per condition).

C   MC1 dot blot with native LUHMES cell protein extract, 1.2 μg protein per dot.

D   Densitometric analysis of Western blots described in (A) (*n* = 3 per condition).

E, F   Reverse transcription quantitative PCR (RT–qPCR) for mRNAs of total tau, 4R tau, and 3R tau (E) and of splicing factors implicated in the shift between 3R and 4R tau isoforms (F) in cells treated as in (A) (*n* = 3). Annonacin specifically increased 4R tau and SRSF2 mRNA expression, and PA reversed this effect.

G   Representative Western blots of protein extracts from LUHMES neurons after lentivirus-induced overexpression of 4R tau in the absence or presence of PA (200 nM). Transgenic 2N4R tau runs at 70 kDa, endogenous tau at 64 kDa.

H   Densitometric analysis of Western blots shown in (G) (*n* = 3).

Data information: Data are mean + SEM, normalized to untreated controls set as 1 (dashed lines). Statistical analysis in (B, H) was by one-way ANOVA with Tukey's *post hoc* test, in (D–F) two-way ANOVA with Tukey's *post hoc* test; *$P < 0.05$, **$P < 0.01$, ***$P < 0.001$ versus control; #$P < 0.05$, ##$P < 0.01$, ###$P < 0.001$, ns: not significant.

These results showed that peripheral administration of the PERK activator is safe, well tolerated, and engages its molecular target in the brains of mice. Therefore, we continued to test its therapeutic efficacy *in vivo*.

## PERK activator decreases pathological tau species *in vivo*

P301S tau transgenic mice were injected i.p. with the PERK activator at 2 mg/kg/day for 6 weeks starting at 17 weeks of age. Again, no observable adverse events occurred. We analyzed tau pathology in the hippocampus, since pathological changes are most pronounced in this region of P301S mice (Xu *et al*, 2014). Optical densities of tau-immunostained sections using HT7 (total tau), MC1 (pathological conformation tau), CP13 and AT180 (phospho-tau) antibodies were significantly decreased upon PERK activator treatment. NeuN staining was unchanged, suggesting that treatment was not neurotoxic (Fig 4C and D). A decrease in all tested epitopes of soluble tau was evident by Western blot analysis (Fig 4E and F). Hyperphosphorylated tau species (detected by CP13) were also decreased in the sarkosyl-insoluble tau fraction (Fig 4G).

## PERK activation protects against memory deficits and dendritic spine loss *in vivo*

We tested visuo-spatial memory deficits which characteristically occur in young P301S tau mice (Xu *et al*, 2014). Therefore, wild-type and P301S tau transgenic mice were treated with saline or PERK activator from week 9 to 15 of age.

We used the Morris water maze (MWM) to measure improvement in the time required to find a hidden platform in a water basin

The mutation at position 301 of the tau protein in this mouse model leads to an early onset memory deficit starting at 10 weeks of age along with the appearance of conformationally changed tau, phosphorylated tau, and dendritic spine loss in the hippocampus. At 20 weeks of age, the mouse develops locomotor deficits correlated with emergence of paired helical filament tau and α-motoneuron loss (Xu *et al*, 2014).

We tested the PERK activator's target engagement *in vivo* by injecting wild-type mice i.p. with different doses for 3 consecutive days. Western blots of brain extracts showed an increase in both pPERK and pNRF2 at a lowest effective dose of PERK activator being 2 mg/kg/day (Appendix Fig S3A and B).

We then treated wild-type mice with the PERK activator i.p. at 2 mg/kg/day for 6 weeks starting at 9 weeks of age. No adverse effects were noted: blood tests for liver and kidney functions were normal; there were no indications of illness and no premature deaths occurring. Western blots of brain extracts showed an increase in both pPERK and pNRF2, but not in total PERK and NRF2 (Fig 4A and B).

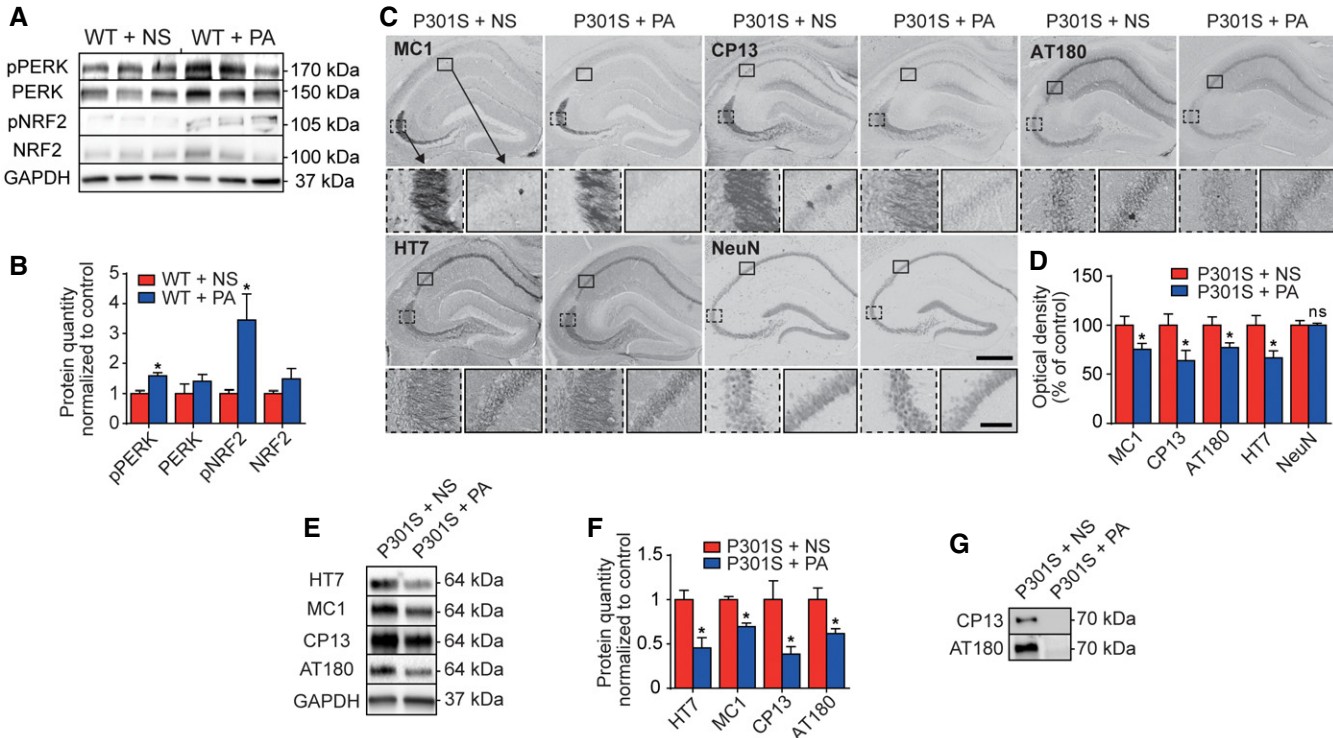

**Figure 4. PERK activator decreases pathological tau species *in vivo*.**

A   Wild-type (WT) mice were treated i.p. with normal saline (NS) as control or PERK activator (PA, 2 mg/kg) once daily for 6 weeks. Western blots prepared from brain homogenates. GAPDH was used as loading control.

B   Densitometric analysis of Western blots described in (A), normalized to WT + NS (*n* = 3 per condition).

C   Representative photomicrographs of hippocampi from 23-week-old P301S tau transgenic mice, treated as described in (A). Sections were immunostained with antibodies against conformationally changed tau (MC1), phosphorylated tau (CP13, AT180), total tau (HT7), and neuronal nuclei (NeuN). Scale bars, upper: 500 μm, lower: 50 μm.

D   Optical density measurement of the CA1/CA2 and CA3 region, as shown in (C) (*n* = 6). Analysis is based on DAB stainings, since this avoids bias by secondary bleaching due to differential storage temperatures/light exposures seen with immunofluorescence.

E   Representative Western blots of soluble protein fractions from whole-brain homogenates of mice, treated as in (C).

F   Densitometric analysis of Western blots as shown in (E), normalized to P301S + NS (*n* = 3).

G   Representative Western blots of sarkosyl-insoluble protein fractions extracted from brain homogenates as in (E), with CP13 and AT180 antibodies.

Data information: Data are mean + SEM. Statistical analysis in (B, D, F) was Student's *t*-test; *$P < 0.05$ versus control, ns: not significant.
Source data are available online for this figure.

after 3 days of learning. Wild-type mice improved drastically during the 3 days of learning (Fig 5A) while P301S transgenic mice improved less, indicating memory impairment. P301S transgenic mice treated with PERK activator performed significantly better.

As structural correlate of memory impairment (Xu *et al*, 2014), we analyzed the morphology of dendritic spines in the CA1 region of the hippocampus (Fig 5B and C). Wild-type mice showed a typical mushroom type of dentritic spines. Their density was reduced in P301S mice. PERK activator treatment restored this deficit partially.

**PERK activation prevents locomotor deficit and motoneuron loss *in vivo***

We finally tested locomotor deficits which characteristically occur in aged P301S tau mice (Xu *et al*, 2014). Wild-type and P301S tau transgenic mice were treated with saline or PERK activator from week 17 to 22 of age.

We used the rotarod test to measure the time span until mice fall off a rotating rod. The time span was significantly reduced in saline-treated P301S tau mice compared to wild-type mice, and restored upon PERK activator treatment (Fig 5D).

As structural correlate of locomotor deficits, we analyzed the degeneration of α-motoneurons. Consistent with prior studies (Allen *et al*, 2002; Scattoni *et al*, 2010), we found swollen and atrophic motoneurons and reduced numbers of intact motoneurons in P301S mice (Fig 5E and F). There was a loss of adductor and quadriceps motoneurons of lamina 9 (Ad9) and lamina Q9, and PERK activator treatment prevented this loss (Fig 5E and F).

# Discussion

Our prior work has demonstrated a stringent genetic association of PERK with the tauopathy PSP (Höglinger *et al*, 2011). In the present study, we analyzed the functional effects of PERK signaling and

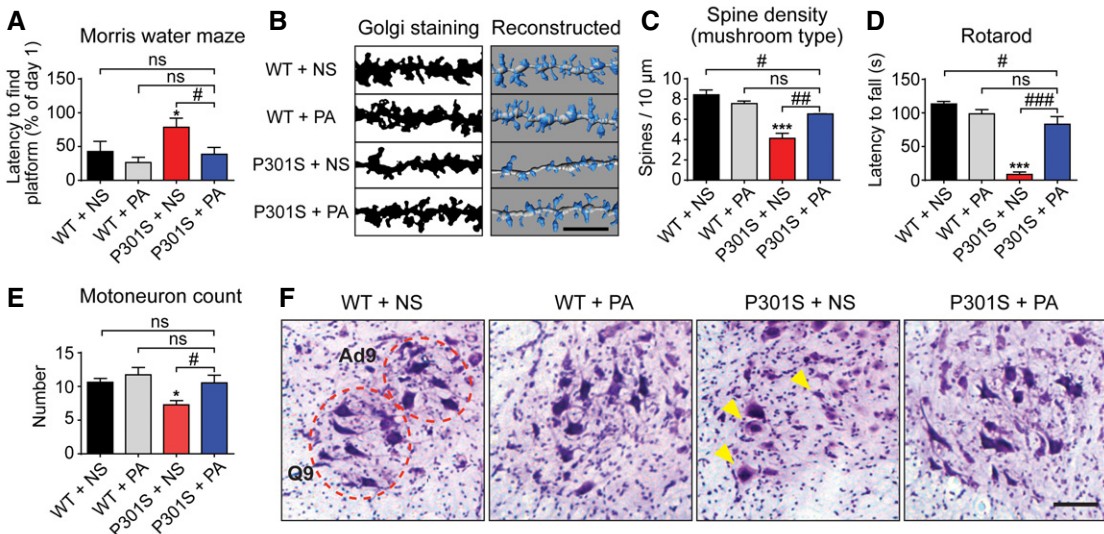

**Figure 5. PERK activator reduces behavioral deficits and neurodegeneration *in vivo*.**

A–C    Young 9-week-old wild-type (WT) and P301S tau transgenic mice were daily injected i.p. with 2 mg/kg PERK activator (PA) or normal saline (NS) for 6 weeks (n = 9 for WT groups; n = 8 for P301S groups). (A) Determination of spatial learning capacity by Morris water maze (MWM). The time required to find a hidden platform in a water basin after 3 training days was compared to day 1. (B) Representative photomicrographs of dendritic spines in the hippocampal CA1 area. Scale bar: 5 μm. (C) Quantification of dendritic spines (n = 3 per condition).

D–F    Aged 17-week-old WT and P301S mice were daily injected i.p. with 2 mg/kg PA or NS for 6 weeks (n = 12 for PA-treated P301S mice; n = 9 for all other groups). (D) Rotarod test was used to evaluate balancing abilities, measuring the time until mice fell off a rotating rod. (E) The number of adductor motoneurons of lamina 9 (Ad9) and quadriceps motoneurons of lamina 9 (Q9) were quantified in lumbar segments 2–3 of the spinal cord on five consecutive sections. (F) Representative photomicrographs of Ad9 and Q9. Arrows highlight examples of abnormally swollen or atrophic neurons. Scale bar: 100 μm.

Data information: Data are mean + SEM. Statistical analysis in (A, C–E) was one-way ANOVA with Tukey's or Fisher's LSD (E) *post hoc* test; *$P < 0.05$, ***$P < 0.001$ versus WT+NS; #$P < 0.05$, ##$P < 0.01$, ###$P < 0.001$, ns: not significant.

provided first evidence for the potential of pharmacological PERK activation in the treatment of tauopathies such as PSP.

Classically, PERK is activated in response to ER stress and then activates NRF2, which protects cells from oxidative damage, and EIF2A, which in turn suppresses protein translation and activates ATF4 (Fig 6A). Surprisingly, our results show that the amount of EIF2A and pEIF2A protein is lower in the frontal cortex of PSP patients compared to controls. Conflicting our findings, Stutzbach and colleagues have reported increased pEIF2A in PSP brains by means of immunohistochemistry using a different antibody (Stutzbach *et al*, 2013). However, consistent with our findings, Unterberger and colleagues have reported increased pPERK but absent pEIF2A in brains of patients with PSP and CBD, two prototypical 4R tauopathies (Unterberger *et al*, 2006). Most importantly, we were able to reproduce pEIF2A suppression by 4R tau overexpression in human neurons. Thus, it is possible that 4R tau mediates the EIF2A and pEIF2A suppression we observed in PSP patients' brains. The exact mechanism, however, requires further investigations.

One potential interpretation of the increased PERK expression in PSP patients is that of an attempt to compensate the effects of pEIF2A suppression through a long-term feedback mechanism. Consistently, we found pEIF2A downregulation already at 2 months of age, but a compensatory PERK upregulation only at 6 months of age in brain homogenates of P301S mice (Fig EV1). However, this effect was not seen in the short-term cell culture models. The resulting activity pattern of the UPR in PSP brains is illustrated in Fig 6B.

In the 3R/4R tau overexpression model, we detected no MC1 signal by Western blot. Thus, overexpression of either isoform is not sufficient to induce conformational change. Only annonacin-treated LUHMES cells showed MC1 tau immunoreactivity. 3R tau is by far more abundant than 4R tau in LUHMES cells. Furthermore, the MC1 signal in annonacin-treated LUHMES cells was running at a molecular weight compatible with the longest 3R tau isoform. Thus, it appears that also 3R tau is MC1-positive.

In our disease models of tauopathy, we observed a much more consistent neuroprotective effect with the PERK activator than with the inhibitor (Fig 6C). This may appear not to directly reflect the results described, for example by Radford *et al* (2015). However, Radford *et al* did not perform a direct comparison of PERK inhibition versus activation. A factor for the greater effect of PERK inhibition may be that Radford *et al* conducted their experiments at a later disease stage in P301L tau transgenic mice when EIF2A levels are higher than in earlier stages of the disease (when their results show reduced EIF2A levels). In addition, we saw the protective NRF2 axis to have greater impact than the EIF2A axis, which explains why we saw a greater effect with PERK activation.

*In vitro* models of environmentally and genetically determined tauopathies were used to explore the protective effects of the PERK activator as opposed to an inhibitor, and to explore implicated molecular mechanisms. The mitochondrial complex I inhibitor annonacin induces a PSP-like tauopathy through mechanisms elucidated in Escobar-Khondiker *et al* (2007). Our *in vitro* models demonstrated that PERK activation reduces phosphorylated and

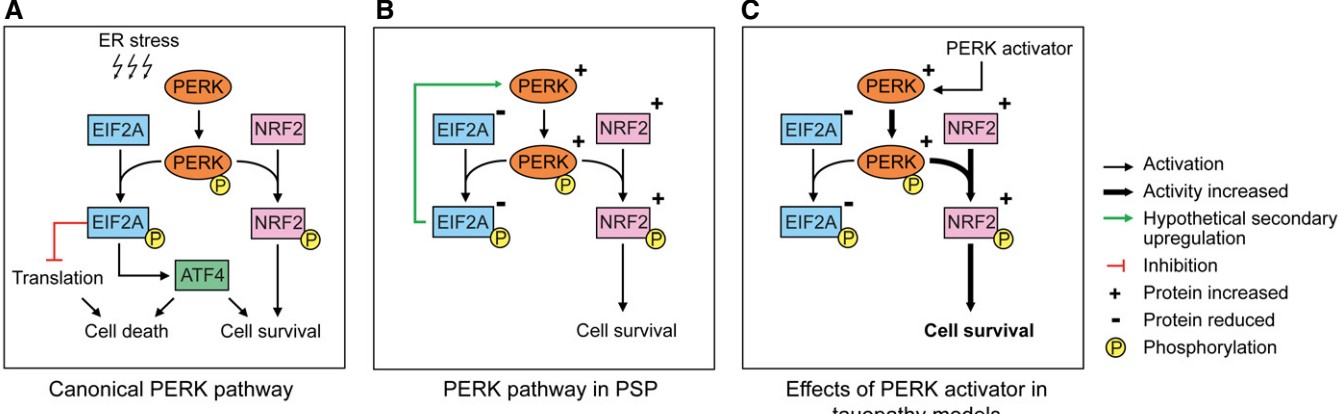

**Figure 6. PERK pathways.**

A  Canonical PERK pathway. An overload of misfolded proteins in the endoplasmic reticulum (ER stress) induces activation of PERK by autophosphorylation. Activated PERK (pPERK) phosphorylates EIF2A (pEIF2A) and NRF2 (pNRF2). pEIF2A leads to total suppression of translation and induces ATF4. Depending on the cellular state, ATF4-regulated transcription may either induce apoptosis, autophagy, or cell survival. pNRF2 induces mechanisms protecting against oxidative stress and promoting cell survival.

B  PERK pathway in PSP. In PSP brains, total EIF2A and pEIF2A protein is decreased. Inversely, total and phosphorylated PERK and NRF2 are upregulated which may be an attempt to compensate for reduced EIF2A levels.

C  Effects of PERK activator in tauopathy models. Similar to PSP brains, amounts of total EIF2A and of pEIF2A are decreased in the corresponding cell models. The PERK activator induces phosphorylation of NRF2. This in turn increases cell survival.

conformationally altered tau, decreases 4R tau isoforms, and protects against neuronal cell death. Our *in vivo* results showed that treatment of mice with the PERK activator CCT020312 leads to increased levels of phosphorylated PERK and NRF2 in brain homogenates (Appendix Fig S3). Therefore, the PERK activator seems capable of penetrating into the brain and engaging with its molecular target. It is also effective in reducing pathological tau species, such as MC1-, CP13-, and AT180-positive tau in both soluble and sarkosyl-insoluble fractions. Intraperitoneal administration of the PERK activator once daily improves the performance of P301S tau transgenic mice in a spatial memory task and in a locomotor task, almost to levels seen in wild-type mice. We observed a rescue of dendritic spine loss in the hippocampus and a rescue of motoneuron loss in the spinal cord in the PERK activator-treated P301S tau transgenic mice (Fig 5A and B). This suggests that both at the functional and cellular level, PERK activation mitigates the detrimental effects of tauopathy. We used human neurons and wild-type tau in our *in vitro* experiments since they are a better proxy for conditions in humans than are mouse models.

This interpretation is consistent with several previous reports of increased UPR activity to protect cells from various forms of protein aggregation (Boyce *et al*, 2005; Loewen & Feany, 2010; Vaccaro *et al*, 2013). It is also compatible with our observation that loss of functional PERK leads to signs of neurodegeneration, including tau-positive neurofibrillary tangles in human brains (Bruch *et al*, 2015). In contrast to these findings, however, a recent study administered a PERK inhibitor to P301L tau transgenic mice suggesting that PERK inhibition may be a viable treatment strategy for tauopathies (Radford *et al*, 2015). While we also observed a small protective effect on PERK inhibition (Fig 2D), this approach contains two key issues. First, PERK inhibition may lead to the same symptoms as Wolcott–Rallison syndrome (Bruch *et al*, 2015), and indeed the treated mice

developed profound pancreatic toxicity. Second, the prion model (Moreno *et al*, 2012, 2013), as well as several other models used previously to show a protective effect of PERK inhibition, also had high levels of pEIF2A. This, however, does not represent the conditions that we observed in human PSP patients, in human cells with annonacin treatment or 4R tau overexpression, or in P301S tau transgenic mice at earlier stages of pathology, which were characterized by low levels of EIF2A or pEIF2A.

In the context of an inhibited EIF2A response, NRF2 appears to become preferentially activated by PERK (Fig 6C). NRF2 is a transcription factor that has been shown to support cell survival especially under conditions of oxidative stress (Cullinan *et al*, 2003). Recently, NRF2 activation has been demonstrated to potently protect against tau pathology (Jo *et al*, 2014). Thus, a possible mechanism for the protective effect of PERK activation may be the predominant NRF2 response.

In summary, our data provide a novel functional model (Fig 6) to explain the association of PERK with the tauopathy PSP (Höglinger *et al*, 2011) and suggest that PERK activation may be a viable strategy to treat PSP and eventually other tauopathies.

## Materials and Methods

### Mice

All animal work was conducted either on male C57BL/6 mice (wild type), obtained from Charles River Laboratories, Wilmington, MA, USA, or on male homozygous transgenic mice overexpressing human tau with the P301S mutation on a C57BL/6 background, originally developed by Michel Goedert (University of Cambridge, UK (Allen *et al*, 2002), obtained by own breeding and their

wild-type littermates. Animals were kept at 23°C ± 1°C under standard 12-h light–dark cycle with free access to food and water. They were handled according to the EU Council Directive 2010/63/EU, the Guide for the Care and Use of Laboratory Animals (National Research Council 2011), and the guidelines of the local institutional committee. The experiments were approved by the local authority "Regierung von Oberbayern" under application number 55.2-1-54-2532-165-13.

### Cell culture

Nunc™ Nunclon™ Delta 6-well (for protein and mRNA) or 48-well (for cell assays) plates (Thermo Fisher Scientific, Waltham, MA, USA) were coated with 100 μg/ml poly-L-lysine (Sigma-Aldrich, St. Louis, MO, USA) and 5 μg/ml fibronectin (Sigma-Aldrich). LUHMES (Lund human mesencephalic) cells, derived from female human embryonic ventral mesencephalic cells by conditional immortalization (Tet-off v-myc overexpression), were seeded out in a concentration of 130,000 cells/cm$^2$ to achieve a confluence of 50%. They were then differentiated for 8 days in a medium of DMEM/F12 (Sigma-Aldrich), 1 μg/ml tetracycline, 2 mg/ml GDNF, and 490 μg/ml dbcAMP into post-mitotic neurons (Scholz *et al*, 2011) with a dopaminergic phenotype (Lotharius *et al*, 2005).

### Pharmacological treatments

Appendix Table S1 gives detailed information about substances used. For all experiments with annonacin and the respective controls, the medium was replaced during the intoxication period with new medium containing glucose levels reduced to 250 μM, that is, the physiological concentration in the human brain (Silver & Erecinska, 1994).

### Cloning of transfer vector plasmids

Plasmids were obtained as indicated in Appendix Table S2. The *MAPT* isoforms *2N3R* and *2N4R* and *EIF2AK3* (the gene encoding mouse PERK) were amplified from the original plasmids by PCR with Q5® High-Fidelity DNA polymerase (New England Biolabs, Ipswich, MA, USA) using the primers as indicated in Appendix Table S3. *MAPT* sequences were inserted into the lentiviral vector plasmid FU-ΔZeo by digesting with the restriction enzyme indicated in Appendix Table S3, followed by ligation with T4 ligase (New England Biolabs). *EIF2AK3* was inserted using the Gibson method (Gibson *et al*, 2009, 2010) with the Gibson Assembly® Cloning Kit (New England Biolabs). The resulting ligated clones were transformed into NEB® 5-alpha competent *E. coli* (New England Biolabs) and grown on an ampicillin-selective Luria Bertani (LB) agar (Sigma-Aldrich) plate. The next day, colonies were picked and amplified in LB medium (Sigma-Aldrich) for 8 h with ampicillin for selection. The plasmid DNA was extracted using the NucleoBond® Xtra Midi Kit (Macherey-Nagel, Düren, Germany) and sequenced for confirmation.

### Lentivirus expression and concentration

As described in Kuhn *et al* (2010), lentiviruses were generated in HEK293T cells. The plasmids psPAX2, pCDNA3.1 (−)-VSV-G, and

the transfer vector FU-ΔZeo (generated as described above) were co-transfected using Lipofectamine® 2000 and Opti-MEM® (both by Life Technologies, Carlsbad, CA, USA); 24 h post–transfection, the medium was replaced with DMEM + pyruvate + GlutaMAX™ (Life Technologies) + 10 % fetal calf serum (Sigma-Aldrich) + 1× essential amino acids (Life Technologies); 48 h after transfection, the medium was ultra-centrifuged for 2 h at 87,000 *g* in a SW28 rotor (Beckman Coulter, Brea, CA, USA) and the supernatant was discarded. The concentrated lentiviral particle pellet was resuspended in TBS-5 (50 mM Tris, 130 mM NaCl, 10 mM KCl, 5 mM MgCl$_2$, 5 % (m/v) BSA) for 4 h at 4°C before aliquoting and storage at −80°C until use.

### Lentivirus transduction

Concentrated lentiviral particles were titrated by transducing LUHMES cells with different dilutions at 6 h after initiating differentiation. The lowest dilution, at which at least 90 % of cells expressed the transgene (as determined by immunocytochemistry of the cells fixed on day 2 post differentiation), was chosen for subsequent experiments. For these, concentrated lentiviral particles were again added into the medium on day 0, 6 h after initiating differentiation.

### Protein extraction from cells

Proteins were extracted by scraping the cells from the culture plate with Mammalian Protein Extraction Reagent (M-PER, Thermo Fisher Scientific) and 1× cOmplete™ Protease Inhibitor Cocktail Tablets (Hoffmann-La Roche, Basel, Switzerland) and 1× PhosSTOP Phosphatase Inhibitor Cocktail Tablets (Hoffmann-La Roche). For pNRF2 blots, cells were extracted with 2% SDS Tris buffer instead of M-PER in order to obtain whole-cell extracts and not to lose the nuclear fraction. The protein solution was frozen at −80°C, immediately after retrieval, for a minimum of 2 h. Then, the solution was thawed on ice, vortexed, centrifuged at 5,000 *g* for 15 min at 4°C, and the supernatant retrieved. Protein concentrations were determined using the BCA Kit (Thermo Fisher Scientific) by heating the samples at 60°C for 30 min and measuring the absorption on a spectrophotometer (NanoDrop 2000c; Thermo Fisher Scientific).

### Human brain tissue

Fresh frozen tissues from the gyrus frontalis superior of seven PSP patients and six controls without psychiatric or neurodegenerative diseases were obtained from the Center for Neuropathology and Prion Research (University of Munich, Germany; Appendix Table S4). All PSP cases had histologically confirmed PSP with Braak stages of IV or higher and extensive frontal cortex involvement. Prior to death, all donors gave written informed consent according to the Declaration of Helsinki for the use of their brain tissue and medical records for research purposes. This work was approved by the local IRB and ethics committee.

### Protein extraction from human and mouse tissue

The relevant tissues were dissected from the fresh frozen sample on dry ice and suspended in 750 μl of Tissue Protein Extraction

Reagent (T-PER, Thermo Fisher Scientific) and 1× cOmplete™ Protease Inhibitor Cocktail Tablets (Hoffmann-La Roche) and 1× PhosSTOP Phosphatase Inhibitor Cocktail Tablets (Hoffmann-La Roche). The tissue piece was initially ground with a pestle inside a microcentrifuge tube on ice and then homogenized by sonication on ice with a Branson Analog Sonifier 450 (Branson, now Thermo Fisher Scientific), with 10 × 500 ms bursts at intensity level 3. The solution was then centrifuged and treated as described above for cells. For extraction of soluble and sarkosyl-insoluble tau fractions, half hemispheres of mouse brains were used, following the protocol previously reported (Xu *et al*, 2014).

### PERK immunoprecipitation

Two hundred micrograms of protein was diluted to equal concentrations with M-PER lysis buffer (Thermo Fisher Scientific) up to 270 μl. The PERK D11A8 antibody (Cell Signaling Technology, Danvers, MA, USA) was added to a concentration of 1:100 and allowed to bind the antigen for 1 h in a rotating 1.5-ml microcentrifuge tube. Then, 30 μl of protein A Sepharose beads (Sigma-Aldrich; 3 mg/ml) were added to each tube and left rotating overnight at 4°C. The tubes were then centrifuged at 10,000 *g* for 3 min, and the supernatant was discarded. The beads were washed in M-PER three times. Each time, the beads were centrifuged and the supernatant was extracted with a Hamilton syringe (Hamilton, Bonaduz, Switzerland) and discarded. The beads were then mixed with 15 μl 1× Roti®-Load 1 (Carl Roth, Karlsruhe, Germany) and heated at 95°C for 5 min. After another centrifugation step of 10,000 *g* for 1 min, the supernatant was loaded onto a gel with the Hamilton syringe and then blotted as described below.

### Western blotting

Twenty micrograms of protein (unless indicated otherwise) was adjusted to equal concentrations between samples by dilution with M-PER and subsequently heated at 95°C for 5 min with Roti®-Load 1 (Carl Roth). SDS–PAGE was performed using Any kD™ Mini-PROTEAN® TGX™ Gels (Bio-Rad Laboratories, Hercules, CA, USA) in a Tris–glycine running buffer. The protein was blotted onto PVDF membrane (Bio-Rad) at 70 V for 65 min on ice. The membrane was blocked with 1× Roti®-Block solution (Carl Roth) for 1 h and then incubated at 4°C overnight under gentle shaking with the primary antibody (Appendix Table S5) in TBS with 5% BSA (Cell Signaling Technology) and 0.05% Tween (Sigma-Aldrich). The membranes were then washed and incubated with the species-specific HRP-bound secondary antibody (Vector Laboratories, Burlingame, CA, USA) at 1:2,500 in 1× Roti®-Block solution for 2 h, followed by further washing and exposure to Clarity Western ECL Substrate (Bio-Rad) or, in the case of MC1 and 4R tau, to Amersham™ ECL™ Prime (GE Healthcare, Little Chalfont, UK). Chemiluminescence was detected with the Gel Doc™ XR System (Bio-Rad) and analyzed by Image Lab™ software (Bio-Rad). All results are presented relative to the actin expression level. In order to determine whether the signal response with the used antibody concentrations and protein concentrations is indeed linear, we created concentration–signal response curves. Cell lysates were diluted in series from 100 to 12.5 μg and loaded to the SDS–PAGE for immunoblotting with different antibodies. All the quantifications of antibodies fit well the linear response

with the indicated protein amount. We therefore deem the detection method used sufficient, especially as we are only evaluating relative differences in protein levels.

### Dot blots

Sixty micrograms of protein was adjusted to equal volumes of 100 μl with distilled water and 2 μl was transferred onto nitrocellulose membrane (Bio-Rad) and allowed to air-dry. Incubations with primary and secondary antibodies, as well as imaging, were performed as described in the Western blotting section.

### Quantitative PCR

RNA from human tissue samples was extracted by grinding the tissue in liquid nitrogen to a powder and then dissolving it in the RA1 buffer supplied as part of the NucleoSpin® RNA Extraction Kit (Macherey-Nagel) + 1% (v/v) 2-mercaptoethanol (Sigma-Aldrich). RNA from cells was extracted by scraping the cells from the culture plate with RA1 buffer + 1% (v/v) 2-mercaptoethanol. The remaining extraction procedure was according to the manufacturer's instructions for the NucleoSpin® RNA Kit (Macherey-Nagel). RNA concentrations were determined using a spectrophotometer (Nano-Drop 2000c, Thermo Fisher Scientific). The RNA was then transcribed into cDNA with the iScript™ cDNA Synthesis Kit (Bio-Rad) using the manufacturer's instructions. Real-time PCR was performed on the Applied Biosystems® StepOnePlus™ system (Life Technologies) using TaqMan® Universal Master Mix II and TaqMan® primers (both Thermo Fisher Scientific) against total *MAPT*, *MAPT 3R*, *MAPT 4R*, *EIF2A*, *EIF2AK3* and *ATF4*. *POL2A* and *PSCM1* served as reference targets as they were most stably expressed in experimental conditions. All values are relative quantities compared to untreated (control) cells. Analysis was conducted with the Applied Biosystems® StepOnePlus™ (Life Technologies) and Qbase+ (Biogazelle, Zwijnaarde, Belgium) software packages. Absolute quantification was performed by creating a standard curve with plasmids containing either the 2N3R or the 2N4R spliced variant of *MAPT* (Appendix Table S2). The absolute quantity was computed by deriving the relationship between Ct values and absolute quantity with the StepOnePlus™ software (Life Technologies).

### ATP assay

ATP assays were conducted using the ViaLight™ Plus Kit (Lonza, Basel, Switzerland) according to the manufacturer's instructions. Luminescence was measured with the FLUOstar Omega plate reader (BMG Labtech). The data were analyzed using the MARS Data Analysis Software (BMG Labtech, Ortenberg, Germany).

### MTT assay

Thiazolyl blue tetrazolium blue (MTT; Sigma-Aldrich) was dissolved in sterile PBS to a concentration of 5 mg/ml. This stock solution was added to the cells in culture medium to achieve a final concentration of 0.5 mg/ml. The 48-well culture plate was then incubated at 37°C for 1 h, the medium removed completely and frozen at −80°C for 1 h. Subsequently, the plate was thawed, 300 μl DMSO (AppliChem, Darmstadt, Germany) was added per well and

the plate was shaken to ensure complete dissolution of the violet crystals; 100 µl from each well was transferred to a new 96-well plate, and the absorbance was measured with a plate reader (FLUOstar Omega; BMG Labtech) at λ = 590 nm (reference, λ = 630 nm). The data were analyzed using the MARS Data Analysis Software (BMG Labtech).

## Cell microscopy

Cells were plated on µ-Slides 8 Well (ibidi GmbH, Munich, Germany) and treated as described in the sections above. Ten days after the initiation of differentiation, cells were fixed in 4 % formaldehyde (Sigma-Aldrich) for 15 min; 4′,6-diamidino-2-phenyl-indole dihydrochloride (DAPI, Sigma-Aldrich) was added for 5 min at 1 µg/ml in PBS before three times washing with PBS. Microscopy was performed on an inverted microscope (DMI6000B, Leica, Wetzlar, Germany). Viable cells were counted in randomly selected areas of 1,300 × 1,000 µm in four replicates each. Counting was done blinded to the treatment condition and repeated three times. For neurite tracing, the "Simple Neurite Tracer" plugin for ImageJ by Mark Longair was used (http://fiji.sc/Simple_Neurite_Tracer). For neurite network density measurement, optical density was measured using ImageJ and converted to a scale of 0 (no neurites) to 100 (total surface area covered with neurites). Sample names were blinded to the experimenter by random numbers before statistical analysis.

## Mouse blood testing

Following anesthesia prior to sacrifice, blood was withdrawn intracardially with a needle and syringe, transferred to Mono-vette® tubes (Sarstedt, Nümbrecht, Germany) and analyzed the same day for testing standard liver and kidney functions by the department of laboratory medicine (Klinikum rechts der Isar, Munich, Germany).

## PERK activator preparation for *in vivo* experiments

Five milligrams of compound CCT020312 (EMD Millipore, Billerica, MA, USA) was first dissolved in 100 µl of sterile DMSO (AppliChem, Darmstadt, Germany) and then diluted with sterile normal saline (0.9 % m/v NaCl; B. Braun Medical, Melsungen, Germany) to achieve a stock solution of 0.5 mg/ml.

## *In vivo* experimental paradigms

For the short-term target engagement trial with ascending doses, 15-week-old wild-type mice ($n$ = 3 per group) were treated on three consecutive days with intraperitoneal (i.p.) injections of either 1, 2, or 5 mg/kg/day of CCT020312. For the long-term target engagement trial, 9-week-old wild-type mice ($n$ = 3 per group) were treated i.p. for 6 weeks with either 2 mg/kg/day of CCT020312 or equivalent volumes of saline. For immunoblotting of tau species in mouse brains, 17-week-old P301S tau transgenic mice ($n$ = 3) were i.p. injected with 2 mg/kg/day of CCT020312 for 6 weeks and sacrificed by cervical dislocation at 23 weeks of age ($n$ = 3 per group). For the efficacy trial on behavioral deficits at the early disease stage, 9-week-old P301S tau transgenic mice ($n$ = 8 per group) and

wild-type mice ($n$ = 9 per group) received i.p. injections of 2 mg/kg/day CCT020312 or equivalent volumes of saline for 6 weeks. They were analyzed by Morris water maze at 16 weeks of age. After the test, mice were sacrificed and brains were dissected for biochemical analyses, Golgi staining, or histological study. To investigate behavioral deficits in the late disease stage (Xu *et al*, 2014) and for motoneuron quantification, 17-week-old P301S tau transgenic mice and wild-type mice ($n$ = 12 for PERK activator-treated P301S tau transgenic mice, $n$ = 9 for other groups) received injections of 2 mg/kg/day PERK activator or saline for 6 weeks i.p.; at 23 weeks of age, they were analyzed with novel open field test and rotarod test. Subsequently, mice were sacrificed. Brains and spinal cords were dissected for biochemical analysis or histological study. All sacrifices were done 12 h after the last injection.

## Morris water maze

To evaluate the spatial learning and memory capacities, the Morris water maze test was performed at 16 weeks of age ($n$ = 8 for P301S tau transgenic mouse groups, $n$ = 9 for wild-type mouse groups). A cylindrical water basin (diameter 120 cm) was filled to a depth of 31 cm with water dyed with tasteless and odorless non-dairy creamer. Four distinct large signs were placed on the four walls of the room as visual cues. The test procedure consisted of 3 days pre-training to familiarize the mice with the test, and 3 days training, as described in Xu *et al* (2014). During the pre-training period, mice had six test runs with a visible platform (diameter 12 cm) on the first day, followed by six runs each on days 2 and 3 with a platform hidden 1 cm below the water surface. Each run lasted a maximum of 2 min. Mice that did not find the platform were gently guided to the platform and allowed to stay on it for at least 10 s. During pre-training, the position of the platform varied from day to day. During the training period, mice had one 2-min trial per day on three consecutive days to find the hidden platform, which remained in a fixed position; 24 h after the training, a probe trial was done without presence of the hidden platform, wherein the mice were allowed to explore the maze freely within 2 min. Tracks of training and probe trial were recorded with the Viewer2 video tracking system (Biobserve, St. Augustin, Germany).

## Rotarod

To monitor the balancing ability at 23 weeks of age ($n$ = 12 for PERK activator-treated P301S tau transgenic mice, $n$ = 9 for other groups), each mouse was placed on a rod with 30 mm diameter rotating at a constant speed of 10 rpm for a maximum of 2 min, as described before (Xu *et al*, 2014). Mice were given six trials 1 day before the test to familiarize themselves with the task. On the following day, they were given six chances to stay on the rotating rod, recording their latency to fall (TSE Systems, Bad Homburg, Germany). The three best performances among the six trials were used for statistical analysis.

## Immunohistochemistry

Animals were anesthetized by i.p. injections of pentobarbital (100 mg/kg) and transcardially perfused with 0.1 M phosphate buffer for 2 min, and then fixed for 10 min with 4%

paraformaldehyde. Post-fixation of the tissue was done in 4% paraformaldehyde at 4°C for 48 h. Brains and spinal cords were prepared for histological analysis, as described before (Xu *et al*, 2014). Post-fixed brains ($n = 6$ per group) were cut into 30 μm thickness of coronal sections with a cryomicrotome (CM3050 S, Leica). For brain sections, free-floating sections were immunostained with NeuN (made in mouse, 1:1,000, Millipore, Darmstadt, Germany), HT7 (made in mouse, 1:500, Thermo Fisher Scientific, Waltham, MA), AT180 (made in mouse, 1:500, Thermo Fisher Scientific), CP13, MC1 (both generous gifts from Prof. Peter Davies, Department of Pathology, Albert Einstein College of Medicine, NY) antibodies, separately raised against neuronal nuclei, total tau, paired helical filament (PHF) tau phosphorylated at threonine 231, phosphorylated tau at serine 202, and conformationally changed tau, respectively. The immunoreactivity was revealed by biotinylated secondary antibodies (donkey anti-mouse, IgG (H+L), 1:1,000, Jackson ImmunoResearch Europe, Suffolk, UK) and 3,3′-diaminobenzidine (SERVA Electrophoresis, Heidelberg, Germany). Images of five consecutive brain sections were captured using a camera (E-330; Olympus, Tokio, Japan) with the same aperture and exposure time. Optical density of the immunoreactivity of antibodies in the hippocampus was measured using Fiji software (http://fiji.sc/Fiji). Corpus callosum on the section was used as background control. Names of slides were blinded by random number for the experimenter until statistical analysis.

### Nissl staining and motoneuron quantification

Post-fixed spinal cords ($n = 3$ per group) were cut into 30-μm-thick coronal sections with cryomicrotome and mounted immediately on gelatine-coated glass slides. Sections were Nissl-stained to reveal neuronal cells, using the FD Cresyl Violet Solution™ (FD NeuroTechnologies, Columbia, MD, USA), following the manual of the manufacturer. The regions of interest were localized with an atlas (http://mousespinal.brain-map.org). The images of spinal cords were captured by microscope (DMI6000B, Leica). Nissl-positive neurons were counted from adductor motoneurons of lamina 9 (Ad9) and quadriceps motoneurons of lamina 9 (Q9) in lumbar 2–3 region. Five consecutive sections of the spinal cord from each mouse were analyzed. The study was blinded by covering the labels of slides with random numbers until statistical analysis.

### Golgi staining and dendritic spine quantification

The study was done following a previously reported protocol (Xu *et al*, 2014). Briefly, mouse brains were impregnated with GolgiStainTM kit (FD NeuroTechnologies) following the manufacturer's guidelines. Four weeks later, impregnated brains were cut into 200-μm-thick coronal sections and developed on the glass slides. Pictures of dendritic spines were captured using a confocal microscope (DMI6000B+TCS SP5II, Leica) under the transparent light channel. The images were analyzed using Imaris software (Bitplane, Zurich, Switzerland). Five pyramidal neurons in the CA1 region from each mouse brain and three apical dendrites with a distance 30–120 μm from each pyramidal neuron were chosen for quantification. Dendritic spines were first reconstructed into a 3D structure and then quantified automatically by the software. Mushroom-type dendritic spine was defined as max width of spine

**The paper explained**

**Problem**

Progressive supranuclear palsy (PSP) is a devastating neurodegenerative disease resulting from pathological aggregation of the tau protein. The *EIF2AK3* gene encoding the PERK protein was identified as risk factor for PSP. Also, PERK protein was found to be activated in vicinity to aggregated tau protein in PSP brains. However, it was still unclear whether activation of PERK is detrimental or protective for neurons in PSP.

**Results**

The present study describes that EIF2A, a substrate of PERK, is suppressed in human PSP and corresponding models. Therefore, PERK predominantly activated its second main substrate, NRF2, in these models. PERK activation reduced pathological tau species and was neuroprotective in the cell culture models of PSP. The PERK activator also reduced behavioral deficits and histological stigmata of tau pathology in tau transgenic mice.

**Impact**

PERK activation may therefore represent a new treatment strategy for PSP and related tauopathies.

head > mean width of spine neck. Labels of slides were covered by random numbers to blind the experimenter until statistical analysis.

### Statistics

Prism 7 (GraphPad Software, La Jolla, CA, USA) was used for statistical calculations and for creating line and bar graphs. In Western blot results, the optical densities of target proteins were always scaled to the respective loading controls and non-treated controls for statistical analysis. Two datasets were compared by *t*-tests. When there are more than two datasets with two variables, data were generally compared by two-way ANOVAs with Tukey's or Dunnett's *post hoc* test, unless indicated otherwise. Assays with one variable were compared by one-way ANOVAs with Tukey's or LSD *post hoc* test. Data are shown as mean + SEM. $P < 0.05$ was considered significant. *P*-values of comparisons, shown in the figures, are listed in Appendix Table S6.

**Expanded View** for this article is available online.

### Acknowledgements

J.B. was funded by the Bavarian Research Foundation (Bayerische Forschungsstiftung), G.U.H. by the Deutsche Forschungsgemeinschaft (DFG, HO2402/6-2 & Munich Cluster for Systems Neurology SyNergy), S.F.L. by the BMBF (JPND RiMOD-FTD), and the Research Award of the Breuer Foundation. This research program and related results were made possible by the support of the NOMIS Foundation (FTLD project). We thank Sigrid Schwarz for her help with spinal cord analyses and Pierre Champy for providing annonacin.

### Author contributions

JB, HX, TWR, KFW, UM, and GUH developed experiments. JB, P-HK, and SFL designed and produced lentiviruses. JB, HX, and ADA performed and analyzed *in vitro* studies. JB and HX performed the study and analyzed results from animal studies. TA provided human brain samples. JB and HX performed

statistical analysis. GUH developed the study, conceived the experimental plans, and analyzed the data. JB, HX, TWR, and GUH wrote the manuscript. All authors read and edited the manuscript and gave final approval of the manuscript version to be published.

## Conflict of interest

A patent application describing PERK activation as treatment strategy for tauopathies is pending (J.B., T.W.R., G.U.H.).

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
