## [Review Process File · EMBO Molecular Medicine]

Manuscript EMM-2016-06664

PERK activation mitigates tau pathology in vitro and in vivo

Julius Bruch, Hong Xu, Thomas W Rösler, Anderson De Andrade, Peer-Hendrik Kuhn, Stefan F Lichtenthaler, Thomas Arzberger, Konstanze F Winklhofer, Ulrich Müller, Günter Höglinger

Corresponding author: Günter Höglinger, German Centre for Neurodegenerative Diseases

Review timeline:

Submission date:	06 June 2016
Editorial Decision:	25 July 2016
Revision received:	24 October 2016
Editorial Decision:	25 November 2016
Additional Correspondence:	25 November 2016
Editorial Decision:	08 December 2016
Revision received:	22 December 2016
Accepted:	23 December 2016

Transaction Report:

Editor: Celine Carret

1st Editorial Decision

25 July 2016

Thank you for the submission of your manuscript to EMBO Molecular Medicine. We have now heard back from the three referees whom we asked to evaluate your manuscript. Although the referees find the study to be of potential interest, they also raise a number of concerns that must be addressed in the next final version of your article.

You will see from the comments pasted below, that the three referees clearly found the study of interest and conclusive enough to be published. However, suggestions to clarify some part of the work (including in the discussion section by refocusing arguments), and strengthen the mechanism (see referees 2 and 3 comments) are proposed to further improve the study. We do agree that such additions would make the paper even more compelling and we would like to invite you to revise your article following these lines.

Please note that it is EMBO Molecular Medicine policy to allow only a single round of revision and that, as acceptance or rejection of the manuscript will depend on another round of review, your responses should be as complete as possible.

Please read below for important editorial formatting.

I look forward to receiving your revised manuscript.

REFEREE REPORTS

Referee #1 (Remarks):

The manuscript on PERK activation mitigates tau pathology in vitro and in vivo describes the mechanistic role of PERK in PSP degenerative diseases. PERK, a RNA-like endoplasmic reticulum kinase is genetically associated with the tauopathy progressive supranuclear palsy (PSP). Authors have studied PERK activity in brains of PSP patients and its function in three tauopathy models (cultured human neurons overexpressing 4-repeat wild type tau or treated with the environmental neurotoxin, annonacin and P301S-tau transgenic mice). On the basis of their large number of experimental findings they conclude PERK activation may be a novel strategy to treat PSP and eventually other tauopathies related to other neurodegenerative diseases. The experiments are conducted systematically and presented nicely, however, there are many controversial results which need explanations. Following are some minor comments.

1. Authors should explain how these contradicting data can be explained.
2. Is PSP pathology restricted to only 4- repeat tau isoform? and why?
3. In AD and other tauopathies, more than one tau isoform is hyperphosphorylated. Now there are more than 42 Tau residues hyperphosphorylated.
4. How conformational changes were measured?
5. Do the Tau conformational changes occur only in 4-repeat Tau?
6. What is the status of other Tau-isoforms in PSP pathology?
7. Is only S-301 residue in Tau is phosphorylated?
8. There are other neuron specific proteins in which pro- directed Ser/Thr residues (ProSer/Thr) ,Neurofilament proteins , NF-M/H , what is the status of their phosphorylation upon neurotoxin , annonacin, treatment of human neurons?
9. Figure representing signal transduction pathways should be simplified.

Referee #2 (Comments on Novelty/Model System):

see below

Referee #2 (Remarks):

This is an interesting study on the beneficial effects of a novel PERK activator. The authors show that activation of PERK is beneficial in two in vitro models of PSP; LUHMES cells expressing 4R tau and annonacin treatment. In addition, the authors show that PERK activation also alleviates the pathology seen in the P301S tau transgenic mouse model.

The paper could be improved significantly by adding some experiments dealing with oxidative stress and thereby providing more mechanistic insight. Annonacin is a blocker of the mitochondrial complex 1 giving rise to high oxidative stress. On the other hand, NRF2 (downstream from PERK) is a strong inducer of heme oxygenase 1 (HO-1), a protein known to be protective against oxidative injury. The authors convincingly show that activation of PERK leads to activation of NRF2. Therefore, the next logical step would be to confirm that HO-1 is indeed up-regulated as a result of this. This could be done on the mRNA or protein level. To check if the positive effects of PERK activation still persist after knockdown of HO-1 would provide convincing insights into the mechanism by which PERK activation could alleviate toxic insults. This could be achieved by co-transfection of cells with PERK and siRNA against HO-1.

In the Results section the following points should be addressed:

The levels of human (p)PERK, (p)NRF2 and (p)EIF2A, especially in their phosphorylated states, should be discussed with more caution because of the long post-mortem delay which accompanies the preparation of human brain tissue. This changes the balance between phosphatase and kinase activities in an artefactual way.

The authors should discuss the toxic mechanism of annonacin (blocking mitochondrial complex I).

One caveat for the interpretation is that the somatodendritic redistribution of hyperphosphorylated tau and cell death can be a hallmark for different types of toxicity, not necessarily tauopathy in the strict sense. Therefore the links to tauopathies should be explained better. For example, tau overexpression and exposure to annonacin appear to be unrelated and trigger different reactions, yet both may lead to tauopathy in humans.

Lentiviral transduction of wild-type 2N4R tau leads to levels 60 times of that in controls, and lentiviral transduction of wild-type 2N3R tau leads to levels 4 times of that in controls, but both of these conditions are not really physiological.

Tau seems to act on EIF2A directly independently of any action on PERK. The mechanism how this may work should be explained better.

Later in the study the authors use P301S tau transgenic mice, but the experiments with cell cultures are done with human LUHMES cells. It would be more consistent to do these experiments in a primary neuronal cell culture of these P301S mice instead of the human LUHMES neurons.

All in all, the second and third paragraphs is somewhat confusing with two seemingly unrelated parameters/toxic insults (Tau overexpression and annonacin exposure) being forced together. Still, the authors show that PERK activation is beneficial in both models.

In paragraph 4-6 of the Results section the authors show that PERK activation protects from annonacin treatment and 4R tau toxicity, and they show that PERK activation reduces tau phosphorylation and prevents it to become folded in a pathological conformation. Additionally, the tau isoform shift is normalized after PERK activation. This data has been confirmed by overexpressing PERK. This makes a strong case that PERK activation is beneficial in both cases.

In vivo, PERK activation reduces total Tau. Consequently, phosphorylated tau and misfolded tau levels are also reduced. Because of this, total tau levels are more important when compared to the other results and therefore should be listed first in the bar graph and emphasized more.

For the discussion section: NRF2 signalling seems to be more important (and more consistent) than EIF2a signaling. Consequently, NRF2 should be given more weight at the expense of EIF2a in the discussion section.

Referee #3 (Comments on Novelty/Model System):

The medium technical quality reflects the use of inferior methods for quantitative western blotting. The medium novelty rating is because PERK is already known to be a risk factor for PSP.

Referee #3 (Remarks):

This manuscript builds on prior evidence that a common variant of the gene encoding RNA-like endoplasmic reticulum kinase (PERK) is a risk factor for progressive supranuclear palsy (PSP), a tauopathy with predominantly 4R tau pathology. The canonical PERK pathway is activated by excess unfolded protein in the ER, and leads to PERK-catalyzed phosphorylation of EIF2A, which then globally suppresses protein translation, and of NRF2, which then activates transcription of mRNA for cytoprotective factors. Tau accumulation was also shown previously to activate PERK. Based on this background, the authors sought to establish whether PERK activation promotes or

protects against PSP, and by extension, to uncover possible PSP therapies based on targeting PERK.

The story begins with analysis of human brain tissue from PSP patients and normal controls. Western blotting demonstrates much higher levels of total and phosphorylated PERK and NRF2 in PSP versus control brains. In contrast, the levels of total and phosphorylated EIF2A were slightly, but statistically significantly lower in the PSP brains compared to the controls. The other most important data concern PERK manipulation by pharmacological and genetic approaches. PERK activator treatment of LUHMES cells, a line derived from mesencephalic neurons, reduced abnormally phosphorylated, tau, conformationally misfolded tau and expression of transfected 4R tau, and protected against cell death induced by annonacin or 4R tau overexpression. Using mice that overexpress pathogenic human P301S tau, the authors also obtained evidence that PERK activator treatment protects against progressive memory, motor function and spinal cord motor neuron loss. In light of these collective findings, the authors propose that activated PERK, working predominantly through NRF2, protects against PSP by somehow reducing levels of toxic 4R tau species. Moreover, they discuss the possible use of PERK-activating drugs as therapeutic agents for PSP.

This is an intriguing study with much merit. It makes a strong case for PERK activation, rather than inhibition, as being protective against PSP, and suggests that the PERK variants that are PSP risk factors are somehow defective in terms of activation or substrate specificity. On the other hand, experimental attention to the PERK risk factor variants, and to genetic manipulation of EIF2A and NRF2 (overexpression and underexpression) could provide much stronger support for the PERK-NRF2 model that the authors favor, and dramatically enhance the value and impact of the study. The data presented are consistent with the PERK-NRF2, as opposed to the PERK-EIF2A model being relevant to PSP, but they are far from definitive. Likewise, they do not inform about why certain PERK variants are risk factors for PSP. In the absence of such data and with the caveats that follow, the current version of paper is certainly worthy of publication now, but might be better suited for a specialty journal.

The authors deserve much credit for their attention to quantitation of results, but they have relied extensively on inferior detection methods to produce critical raw data. Most importantly, the frequently used quantitative western blotting is based on chemiluminescence, which has a linear response range of ~1 order of magnitude, but there is no evidence that any of the quantitation was performed within linear response ranges. This does not compromise the qualitative interpretation of the blots, but it does compromise the quantitation. For their future work, the authors are strongly urged to use infrared fluorescent detection, like that provided by LiCor Odyssey or GE Healthcare Typhoon imaging stations, which are more sensitive than enhanced chemiluminescence systems and have a linear response range of 4-6 orders of magnitude. The quantitation of immunoperoxidase-labeled brain sections in figure 4C is of similarly limited utility. Again, immunofluorescence detection would permit much more reliable quantitation.

Additional minor issues that warrant attention include the following. 1) Define LV-mCh in the legends for figures in which the acronym is used (1C, for example). 2) Define "chromatin clumps" for figure 2E,F, and what is the justification for stating that the clumps mark dead neurons? 3) The value of figure 2F would be improved if it were supplemented by quantitation of neurite density using multiple randomly chosen fields of view for each condition. 4) There is an anomaly in figure 3: how can total and 3R tau protein and mRNA not be affected by annonacin, while 4R protein and mRNA increases substantially? 5) On lines 194-195, the sentence "P301S transgenic mice treated with PERK activator performed significantly" seems to be missing a final word (better?).

1st Revision - authors' response

24 October 2016

Referee #1

1. Authors should explain how this contradicting data can be explained

Response: We have now elaborated on our explanation of any apparent conflicts in results between our results and the previously published results supporting PERK inhibition as a therapeutic rationale (see yellow highlights in manuscript text rows 249-250, 256-258, 271-277).

2. Is PSP pathology restricted to only 4- repeat tau isoform? and why?

Response: We have added a reference to the literature about the different isoform expressions in PSP vs. controls (Chambers et al, 1999, rows 39, 157). The neurofibrillary tangles seen in tau are indeed predominantly of the 4R isoform.

3. In AD and other tauopathies, more than one tau isoform is hyper-phosphorylated. Now there are more than 42 Tau residues hyper-phosphorylated.

Response: The manuscript text now clarifies that we are only detecting individual phosphorylation sites with the antibodies used, but that these can be taken to be representative for wider hyperphosphorylation (see Feany et al, 1995; Wray et al, 2008 as referenced in the manuscript text rows 87-88).

4. How conformational changes were measured?

Response: The conformational changes were detected by Western blot and confirmed by non-denaturing dot-blot with the MC1 antibody, which detects a pathological conformational change in tau protein (see Jicha et al, 1997 or Weaver et al, 2000). We have now clarified this in the manuscript (row 145).

5. Do the Tau conformational changes occur only in 4-repeat Tau?

Response: An immunoprecipitation with MC1 antibody, followed by 3R-Tau vs. 4R-Tau Western was, unfortunately, not doable in the given time frame. Co-immunostaining for 3R-Tau – CM1 and 4R-Tau – CM1 with confocal imaging is not promising. However, we have clarified the following in the manuscript text: In the 3R/ 4R tau overexpression model we observed no MC1 signal in Western blot. Thus, overexpression of either isoform is not sufficient to induce conformational change. Only in annonacin treated LUHMES cells we did observe MC1 immunoreactivity tau. 3R tau in LUHMES cells is by far more abundant than 4R in LUHMES cells. Furthermore, the MC1 signal in annonacin-treated LUHMES cells was running at a molecular weight compatible with the longest 3R tau isoform. Thus, it appears that also 3R Tau is MC1 positive (rows 265-269).

6. What is the status of other Tau-isoforms in PSP pathology?

Response: We describe the status of tau-isoforms in PSP pathology in our previous paper (Bruch et al. PLoS One 2014). We have now explained this and referenced this more explicitly in the manuscript text (rows 157-159).

7. Is only S-301 residue in Tau is phosphorylated?

Response: As described under point 3, many residues in Tau get phosphorylated. The S-301 residue is mutated from a CCG to TCG residue in the mouse model used – unrelated to any phosphorylation at this site. We hope to have made this more explicit in the current manuscript text (e.g. row 196).

8. There are other neuron specific proteins in which pro- directed Ser/Thr residues (ProSer/Thr), Neurofilament proteins, NF-M/H, what is the status of their phosphorylation upon neurotoxin, annonacin, treatment of human neurons?

Response: We have now done a new series of Western blots with phosphorylated neurofilament-medium polypeptide (pNFM) antibody that show pNFM to decline with annonacin treatment, but to be restored with PERK activator treatment (see figure EV6 in the supplementary material).

9. Figure representing signal transduction pathways should be simplified.

Response: Thank you for pointing out its complexity. We have now simplified the figure (see figure 6).

Referee #2:

1. The paper could be improved significantly by adding some experiments dealing with oxidative stress and thereby providing more mechanistic insight. Annonacin is a blocker of the mitochondrial complex 1 giving rise to high oxidative stress. On the other hand, NRF2 (downstream from PERK) is a strong inducer of heme oxygenase 1 (HO-1), a protein known to be protective against oxidative injury. The authors convincingly show that activation of PERK leads to activation of NRF2. Therefore, the next logical step would be to confirm that HO-1 is indeed up-regulated as a result of this. This could be done on the mRNA or protein level. To check if the positive effects of PERK activation still persist after knockdown of HO-1 would provide convincing insights into the mechanism by which PERK activation could alleviate toxic insults. This could be achieved by co-transfection of cells with PERK and siRNA against HO-1.

Response: We have now performed a series of Western blots using an antibody against heme oxygenase-1 (HO-1) showing HO-1 upregulation upon PERK activator treatment (see figure EV5). For technical reasons we were not able to perform a HO-1 knockdown. We also used the NRF2 activator DL-sulforaphane-N-acetyl-L-cysteine (SFN-NAC) and found 4R tau expression to have a less toxic effect on ATP assay with SFN-NAC than without. Correspondingly we saw a more toxic response on knockdown with siRNA targeting NFE2L2, the NRF2 gene. These results demonstrate the significance of NRF2 in protection against 4R tau toxicity.

2. In the Results section the following points should be addressed:

The levels of human (p)PERK, (p)NRF2 and (p)EIF2A, especially in their phosphorylated states, should be discussed with more caution because of the long post-mortem delay which accompanies the preparation of human brain tissue. This changes the balance between phosphatase and kinase activities in an artefactual way.

Response: We have added a section in the results section demanding caution on the interpretation of phosphorylation results in human brain tissue (rows 72-74).

2. The authors should discuss the toxic mechanism of annonacin (blocking mitochondrial complex I).

Response: We have added a discussion of the toxic mechanism of annonacin in the manuscript (rows 81-85 and 280-282).

3. One caveat for the interpretation is that the somatodendritic redistribution of hyperphosphorylated tau and cell death can be a hallmark for different types of toxicity, not necessarily tauopathy in the strict sense. Therefore the links to tauopathies should be explained better. For example, tau overexpression and exposure to annonacin appear to be unrelated and trigger different reactions, yet both may lead to tauopathy in humans.

Response: The link is explained in our prior publication Escobar-Khondiker et al. J. Neuroscience, 2007. We have additionally inserted this reference and discussed its implications in the results section rows 84-85 and 281-282).

4. Lentiviral transduction of wild-type 2N4R tau leads to levels 60 times of that in controls, and lentiviral transduction of wild-type 2N3R tau leads to levels 4 times of that in controls, but both of these conditions are not really physiological.

Response: We have explained the implications of this better in the manuscript (rows 93-95). At the age of harvesting (generally 10 days post differentiation) the LUHMES cells only have a relatively low concentration of 4R tau (which is still a major improvement to other models which generally have none). The lentivirus brings the 4R tau levels to levels roughly equivalent to that of 3R overexpression, although from a much lower base. As in an adult human the levels of 3R and 4R tau are similar, the result is still relatively close to the physiological state, especially as naturally there are 10-fold fluctuations on tau isoform concentration (Mangin et al, 1989).

5. Tau seems to act on EIF2A directly independently of any action on PERK. The mechanism how this may work should be explained better.

Response: Nothing seems to be known about it in the literature and investigating this would be beyond the scope of this paper. We have, however, mentioned in the discussion that this must be investigated further (rows 256-258).

6. Later in the study the authors use P301S tau transgenic mice, but the experiments with cell cultures are done with human LUHMES cells. It would be more consistent to do these experiments in a primary neuronal cell culture of these P301S mice instead of the human LUHMES neurons.

Response: We have now explained better our rationale for using human neurons for the first set of experiments. The tau composition in mouse neurons is different (there is no 4R tau). Also, primary cultures from P301S tau transgenic mice do not display frank cell death vs. wild type cells under reasonable experimental conditions (rows 292-293).

7. All in all, the second and third paragraphs are somewhat confusing with two seemingly unrelated parameters/toxic insults (Tau overexpression and annonacin exposure) being forced together. Still, the authors show that PERK activation is beneficial in both models.

Response: Thank you for pointing this out. We have now modified and clarified this section accordingly.

8. In paragraph 4-6 of the Results section the authors show that PERK activation protects from annonacin treatment and 4R tau toxicity, and they show that PERK activation reduces tau phosphorylation and prevents it to become folded in a pathological conformation. Additionally, the tau isoform shift is normalized after PERK activation. This data has been confirmed by overexpressing PERK. This makes a strong case that PERK activation is beneficial in both cases.

Response: Thank you

9. In vivo, PERK activation reduces total Tau. Consequently, phosphorylated tau and misfolded tau levels are also reduced. Because of this, total tau levels are more important when compared to the other results and therefore should be listed first in the bar graph and emphasized more.

Response: Thank you for pointing this out – we have rearranged the figure accordingly.

10. For the discussion section: NRF2 signaling seems to be more important (and more consistent) than EIF2a signaling. Consequently, NRF2 should be given more weight at the expense of EIF2a in the discussion section.

Response: We have now emphasized NRF2 more in the discussion and our new additional experiments with the NRF2 activator DL-sulforaphane-N-acetyl-L-cysteine (SFN-NAC) and *NFE2L2* siRNA highlight the significance even further (rows 132-142, 276-277). However, we feel the controversy about EIF2A function means it also needs to be discussed in some detail.

Referee #3

1. The medium technical quality reflects the use of inferior methods for quantitative western blotting.

Response: We would like to argue that for the purpose of this paper we were only interested in relative changes of protein concentrations and we therefore believe our method of quantitative Western blotting is sufficient for our purposes. We have described this rationale in the methods section (428-433) and performed additional experiments to show the linearity of the response (see point 4 below).

2. The medium novelty rating is because PERK is already known to be a risk factor for PSP.

Response: PERK is indeed already known to be a risk factor for PSP (as indeed initially described by us), but the molecular mechanisms underlying this purely descriptive association have not been known so far. This is the innovation step of this manuscript. We have made this more explicit in the introduction (rows 61-63).

3. It makes a strong case for PERK activation, rather than inhibition, as being protective against PSP, and suggests that the PERK variants that are PSP risk factors are somehow defective in terms of activation or substrate specificity. On the other hand, experimental attention to the PERK risk factor variants, and to genetic manipulation of EIF2A and NRF2 (overexpression and underexpression) could provide much stronger support for the PERK-NRF2 model that the authors favor, and dramatically enhance the value and impact of the study. The data presented are consistent with the PERK-NRF2, as opposed to the PERK-EIF2A model being relevant to PSP, but they are far from definitive. Likewise, they do not inform about why certain PERK variants are risk factors for PSP. In the absence of such data and with the caveats that follow, the current version of paper is certainly worthy of publication now, but might be better suited for a specialty journal.

Response: We have significantly strengthened the rationale for the PERK-NRF2 axis being critical in PSP. Figure EV5 shows that knockout of the NRF2 gene *NFE2L2* with siRNA amplifies 4R tau toxicity and that activation of NRF2 with DL-sulforaphane-N-acetyl-L-cysteine (SFN-NAC) reduces it. We agree that experiments with overexpression or underexpression of NRF2 and EIF2A and with different risk snps of PERK would be very interesting and definitely should be investigated further. However, such experiments would be beyond the scope of what is doable for this paper.

4. The authors deserve much credit for their attention to quantitation of results, but they have relied extensively on inferior detection methods to produce critical raw data. Most importantly, the frequently used quantitative western blotting is based on chemiluminescence, which has a linear response range of ~1 order of magnitude, but there is no evidence that any of the quantitation was performed within linear response ranges. This does not compromise the qualitative interpretation of the blots, but it does compromise the quantitation. For their future work, the authors are strongly urged to use infrared fluorescent detection, like that provided by LiCor Odyssey or GE Healthcare Typhoon imaging stations, which are more sensitive than enhanced chemiluminescence systems and have a linear response range of 4-6 orders of magnitude.

Response: We would like to thank the reviewers for pointing out a potentially higher quality technique for quantification. However, we believe that the chemiluminescence based technique used was indeed fully sufficient for the experiments at hand, especially as we are only evaluating relative differences in protein levels. As the diagrams below show, the signals are in the linear range for the antibody concentrations and protein loads used. We have added this fact to the methods section now (rows 428-433), including a description of the methodology used for the figure below. For future work, we have now indeed established a LiCor Odyssey.

Fig 1: Cell lysates were diluted in series from 100 to 12.5 µg and loaded to the SDS-PAGE for immunoblotting with different antibodies. All the quantifications of antibodies well fit the linear response with the indicated protein amount.

5. The quantitation of immunoperoxidase-labeled brain sections in figure 4C is of similarly limited utility. Again, immunofluorescence detection would permit much more reliable quantitation.

Response: We have explained our rationale in the figure legends (row 868-870). For the purpose of this experiment, we preferred DAB since this avoids bias by secondary bleaching due to differential storage temperatures / light exposures seen with immunofluorescence.

6. Define LV-mCh in the legends for figures in which the acronym is used (1C, for example).

Response: We have added additional definition and explanation in the figure legends (row 837)

7. Define "chromatin clumps" for figure 2E,F, and what is the justification for stating that the clumps mark dead neurons?

Response: We have explained this in the figure legend (rows 840-842).

8. The value of figure 2F would be improved if it were supplemented by quantitation of neurite density using multiple randomly chosen fields of view for each condition.

Response: We have now followed this suggestion, as shown in supplementary figure EV4. PERK activator is shown to protect against neurite damage by both 4R tau and annonacin.

9. There is an anomaly in figure 3: How can total and 3R tau protein and mRNA not be affected by annonacin, while 4R protein and mRNA increases substantially?

Response: The reason is that 4R tau is present in much lesser concentration in the cells (see results for quantification). Therefore, a significant increase in 4R tau may not lead to a significant increase in total tau. We have now explained this in the figure legend (rows 852-853).

10. On lines 194-195, the sentence "P301S transgenic mice treated with PERK activator performed significantly" seems to be missing a final word (better?).

Response: Thank you for pointing this out. We have corrected this now.

Thank you for the submission of your revised manuscript "PERK activation mitigates tau pathology in vitro and in vivo". We are sorry that it has taken longer than usual to get back to you on your manuscript. We experienced delays in securing the re-evaluations (and unfortunately only one referee agreed to re-review) and I also wished to discuss this case with an external expert, who was not immediately available.

As you will see, while referee 3 acknowledges the effort of addressing some of the issues raised, unfortunately fundamental concerns remain that preclude publication of the manuscript in EMBO Molecular Medicine.

Please rest assured that I took great care of discussing this decision within the team and included one of our expert advisor, who unfortunately, agreed with referee 3 and did not recommend publication of the article in EMBO Molecular Medicine. As you know, we only allow one round of major revision, and for this reason and given these negative evaluations, I do not see any other choice than to return the article to you with the message that we cannot consider it further.

I am truly sorry to have to disappoint you on this occasion, but hope you will find soon a better suited venue for your study.

REFeree REPORT

Referee #3 (Remarks):

As a reviewer of the prior version of this manuscript, I made several suggestions for how to improve the paper. With one exception, all of the suggestions were addressed. The suggestion that was ignored was to overexpress and knock down EIF2A and NERF2. This suggestion was made in light of the evidence from the Radford et al, 2015 paper that PERK activation of EIF2A is neurotoxic and can be ameliorated by a PERK inhibitor, whereas the current study points to activation of PERK as being neuroprotective as a consequence of PERK-mediated phosphorylation of NERF2. Overexpression and knockdown of EIF2A and NERF2 could help to resolve this discrepancy between the two studies. Furthermore, the discordant results between the two studies are complicated additionally by an anomaly in the current paper's data. Whereas the studies with LUHMES cells show clear protective effects of the PERK activator, the starting point for the paper is evidence for upregulation of total and phospho-PERK, and phospho-NERF2 in human PSP brain (Fig 1). Unless I misunderstand something (it would not be the first time!), the human brain and cultured cell data are thereby diametrically opposed to each other.

In my opinion, the authors did respond adequately to some, but by no means all of the reviewers' comments. This seems especially true for the controversy over whether PERK activation promotes or protects against PSP, and whether protection, if it occurs, is mediated by EIF2A, NERF2 or a combination of the two. In light of these issues I cannot recommend publication of the present version of the paper.

We thank you for the re-evaluation of our manuscript.

We are all very surprised that the manuscript was rejected, since we have adequately addressed all doable reviewers' requests.

Reviewer 3 claimed that we ignored his proposal to overexpress and knock down EIF2A and NERF2.

However, we have actually done this:
- siRNA-mediated NRF2 silencing (new Figure EV5).
- NRF2 activation (new Figure EV5).

- EIF2A knock-down: active pEIF2A is significantly downregulated in all our models anyway (Figures 1D, EV2, EV3).
- EIF2A overexpression was the only thing not done, because the generation of an overexpression virus is simply not doable within the given 3 month time frame. Also we wonder why we should overexpress EIF2A, if all our data show that active EIF2A is DOWNregulated in our models. This is one of the core findings, discussed extensively in the paper.

Furthermore referee 3 argues that the human brain and cultured cell data are diametrically opposed to each other.
(Whereas the studies with cells show clear protective effects of the PERK activator, the starting point for the paper is evidence for upregulation of total and phospho-PERK, and phospho-NERF2 in human PSP brain)

However, we tried to make very clear in the discussion, that these data actually match very well: In analogy to endogenous PERK activation plus addition of a PERK activator, a child may partly, yet insufficiently succeed to swim, only the additional application of a swimming aid might prevent it from drowning.

- This reviewer provided medium novelty rating because PERK is already known to be a risk factor for PSP. However it is obvious that the initial purely statistical association of a gene with a disease (Höglinger et al., Nat Genetics, 2011) is only the trigger for the exciting quest to understand the causal relationship (current manuscript). This is true for PERK as it is for tau and alpha-synuclein and amyloid-beta and TREM2 and others.

- He provided medium technical quality ratings due to the 'use of inferior methods for quantitative western blotting', albeit we had used generally accepted standard methodology and now validated all findings with 'superior methodology'.

Please understand that we do not feel our manuscript has received a fair and unbiased review at this occasion.

We would sincerely appreciate you and your team to have a careful second look in light of these arguments.

3rd Editorial Decision

08 December 2016

Thank you for your patience while I sought further advice on your article. I apologise for the delay but I believe it was worth the wait as our adviser recommends publication pending final amendments and details (see below). Our adviser agrees that you have indeed satisfactorily performed the NRF2 KD / activation experiment. It looks like several unusual circumstances happened making it harder for referee 3 to identify the changes. We believe that the main reason is that the figures EV, including the critical EV5, were uploaded as SI, and not as individual figures as requested in our guidelines, meaning that they were not included in the final merge of the article downloadable in one click for referees to see and this could be the reason why this referee missed it. As I was eager to send the article back to referees I didn't think twice about the format, and it turns out this was a mistake and I am sorry.

Anyhow, as I just mentioned, we will be able to accept the manuscript pending the following final changes:

- 1) Please see the adviser's comments and respond adequately in a point-by-point letter (specifically, please provide the full Western Blot showing NRF2 reduced expression following silencing).

Please submit your revised manuscript as soon as possible.

I thank you once more for your patience and cooperation.

***** Advisor's comments *****

In my view the authors have adequately addressed the concerns raised by Ref1 and Ref2. One exception is point 6 of Ref2 where the authors state that mouse tau does not contain 4-repeat isoforms which is clearly not the case. More elaboration on this point is necessary.

The concerns of Ref3 were also addressed, in principle, as described in the response letter by the authors. However, the authors make the claim (in the figure legend of EV5) that a successful knockdown of NRF2 was shown by western blot. This has not been included in the figure but should be done. [...]

Furthermore, I feel that the criticism of Ref3 regarding the chemiluminescence western blot is not justified; this method is suitable to support the claims made by the authors in this manuscript. The referee actually acknowledges that the dynamic range of chemiluminescence is one order of magnitude, which is sufficient to support the claims and data made in this manuscript.

Ref3 holds that the data in human PSP brain (high PERK) contradict the data generated in LUHMES cells, which show that PERK can be protective. I do not share this opinion. Upregulation of PERK may well reflect an effort by the PSP brain to protect itself against the tau-induced damage, and the authors explain sufficiently well how and why PERK activity can be neuroprotective.

3rd Revision - authors' response

22 December 2016

Referee #4:

“However, the authors make the claim (in the figure legend of EV5) that a successful knockdown of NRF2 was shown by Western blot. This has not been included in the figure but should be done. [...]”

Response: We added a new figure (Appendix Fig S4) which shows the Western blots of the silencing efficacy (Appendix Fig S4A) and the quantification of the effect (Appendix Fig S4B).

Corresponding Author Name: Prof. Dr. Günter U. Höglinger

Manuscript Number: EMM-2016-06664